# PVDF Based Pressure Sensor for the Characterisation of the Mechanical Loading during High Explosive Hydro Forming of Metal Plates

**DOI:** 10.3390/s21134429

**Published:** 2021-06-28

**Authors:** Jérémie Tartière, Michel Arrigoni, Alain Nême, Hugo Groeneveld, Sjoerd Van Der Veen

**Affiliations:** 1ENSTA Bretagne, IRDL UMR 6027 CNRS, 2 rue François Verny, 29806 Brest, France; jeremie.tartiere@ensta-bretagne.org (J.T.); alain.neme@ensta-bretagne.fr (A.N.); 23D Metal Forming, Karperweg 8, 8221 RB Lelystad, The Netherlands; hugo.groeneveld@3dmetalforming.com; 3Airbus Operations SAS, 18 rue Marius Terce, 31300 Toulouse, France; sjoerd.van-der-veen@airbus.com

**Keywords:** metal forming, PVDF shock gauge, underwater explosion, shock wave, numerical simulation

## Abstract

High explosive hydro forming (HEHF) is a suitable technique for large metal plate forming. Manufacturing stages of such a part requires an adapted design of explosive charge configurations to define the mechanical loading exerted on the part. This mechanical loading remains challenging to be experimentally determined but necessary for predictive numerical simulation in the design of parts to form. Providing that the actual mechanical impulse would allow the neglecting of the modelling of the detonation stage, this considerably increases the computational time. The present work proposes an experimental method for obtaining the exerted mechanical loading by HEHF on the part to form. It relies on the development of low-cost sensor based on a polyvinyliden fluorid (PVDF) gauge. In addition to it, an analytical approach based on shock physics is proposed for the sensor signal interpretation. The method considers the multi-layer aspect of the sensor and its intrusiveness with respect to waves propagation. Measurements were repeated to assess their relevance and the reproducibility by using steel and aluminium anvils in HEHF. Numerical modelling in 2D plane geometry of the experiments was performed with two commercial hydrocodes. The comparison of mechanical impulses shows an agreement in terms of chronology but a noticeable difference in terms of amplitude, explained by mesh size and numerical diffusion.

## 1. Introduction

The use of high explosives as an available source of energy for metal forming started to be prospected as early as in the middle of the last century [1]. The high explosive hydro forming (HEHF) process consists of holding a metal plate on a die giving the final shape, a vacuum is ensured between the die and the plate. Then an explosive charge is positioned above the plate at a specific distance and the assembly—die and plate and explosive charges—is immersed into a pool. The explosive charge is fired, and the plate is formed due to the mechanical work given by the shock wave resulting from the underwater detonation. In order to obtain the desired final shape, it usually requires time and many trials to find the adapted setting. This iterative process can be time demanding and costly in the case of large metal plates. HEHF has many advantages: the process is not so expensive since the explosive is of a reasonable cost and the forming does not require high cost of maintenance, with respect to maintenance costs of conventional forming processes [2]. Moreover, shapes that cannot be formed by traditional forming processes, can be achieved with large and thick plates by HEHF, especially in aircraft manufacturing [3]. For instance, the front of an aircraft cockpit is currently made of several pieces that are costly and tricky to assemble and the result is not as weight efficient as wished. Those parts could be obtained in one thick and large metal plate using HEHF and machining, even considering their double curvature areas. However, the HEHF process is physically complex and needs to be better understood in order to be adapted for specific productions in the frame of industrial plants. Analytical and numerical approaches are needed for the definition of the explosive charge and the dye.

Furthermore, when developing predictive numerical modelling, engineers are coping with a computation time that is dependant of the complexity of the brief and intense phenomenon to be considered, especially for underwater detonation related studies [4,5]. For instance, high explosive detonation has a characteristic time below the microsecond, then the underwater shock that reaches the plate in few tens of microseconds and creates a fluid–structure interaction that is not always well implemented in numerical codes and considerably increases the computational time [6]. Then, the full forming process that lasts few milliseconds remains a challenge in term of formability limit, and finally the bubble made by the detonation products pulsates and generates new shocks in a few tens to hundreds of milliseconds. Thus, to be realistic, the numerical model would require a very dense mesh and, thus, a very short simulation time step to take into account all those physical effects. That is why correlating the computed pressure with analytical solutions during the interaction between the primary shock and a metal part would allow faster HEHF modelling.

To ensure the validation of the mechanical loading deduced from both numerical and analytical approaches, it is necessary to compare them with experimental measurements resulting from HEHF. In the present work, the explosive charge is a very long cylinder. To the author’s knowledge, no relevant analytical solutions are available, in the open scientific literature, for describing the underwater blast and shock propagation resulting from the detonation of long cylindrical explosive charges. Cole [7] described an underwater explosion for a spherical charge in far field, and his approach is further investigated by Geers and Hunter [8]. Kirkwood and Beth [9] proposed in 1942 a model for plane, spherical, and cylindrical shapes in near field cases. It was completed in 2012 by Barras [6]. A description of an underwater detonation of cylindrical charges in a far field was proposed by Hammond [10], but only for small length to radius ratios (L/R).

An experimental measurement technique of the over-pressure exerted on the plate during HEHF would allow determining of the mechanical impulse acting on the plate to form. This measurement would be considered as an input data in a weak coupling modelling approach, much less costly in terms of computational time.

The aim of this article is thus to propose first a measurement method of the primary shock impulse (maximum pressure and positive phase duration) induced by underwater explosion of detonating cords above the surface of an immersed metal anvil in a HEHF pool. Due to the complexity of the measurement of such interaction, and for experimental reasons, a piezo-electrical polyvinyliden fluorid polymer (PVDF) shock gauge [11] was implemented as a dynamic pressure sensor on the loaded surface of the metal plate to form. This technique has been successively used for field pressure measurement in dry conditions, in aeronautic applications [12] involving low pressure and also in plate impact experiments [13] with much higher pressure. In most papers dealing with dynamic pressure measurement, the method of interpretation from voltage to pressure is most of the time masked. The present paper gives details of sensor implementation and signal interpretation. The second objective of this work is the validation of numerical modelling—of underwater shock wave acting on thick and large metal plate—performed with two different commercial codes—SPEEDv3.1^®^ —(Numerics Gmbh) based on computational fluid dynamic (CFD) and—RADIOSSv2017^®^ —(ALTAIR Hyperworks) that uses a finite element method in an explicit scheme. The third objective is to give analytical interpretations, based on the shock polar technique, of experimental observations carried out on large and thick metal plate. For confidential reasons, the explosive charge and pool details are not evoked. Thus, all time, pressure, and distance results implying the explosive are dimensionless.

## 2. Materials and Methods

### 2.1. PVDF Sensor Description and Implementation

The underwater explosion (UNDEX) pressure measurements are performed in the HEHF pool of the 3DMF company, using PVDF made pressure sensors. Pressure sensors are composed of thin film PVDF (25 μm, Bauer S25 PVDF shock gauge) designed for shock pressure measurement (Figure 1). The reported maximum measured pressure is higher than 30 GPa [14] and the response time is of the order of the nanosecond. They are currently used for blast [15] and shock pressure measurement [16,17], they have been described by F. Bauer [14]: the PVDF semi-crystalline polymer is mechanically stretched and polarised under the effect of a high voltage electric field in order to obtained a calibrated piezo-electric effects described in Figure 2.

The intersection of the two conductive leads (gold or aluminium sputtered on the PVDF polymer) constitutes the measurement zone of the gauge. The use of a gauge having a 25 mm${}^{2}$ measurement surface allows having more electrical charges per surface unit than it is collected with the S9 Bauer shock Gauge presented in [14]. As a consequence, the amplitude of the measured signals and, thus, the accuracy of the measurement is valuable in the case of a planar contact between the shock wave and the active surface of the PVDF gauge. However, for brief mechanical impulses with a very curved front wave, the measurement may remain questionable.

The PVDF gauge is also a broadband antenna that may explain an unwanted triggering during tests due to its sensitivity to electromagnetic and capacitor parasites. Electromagnetic shielding of the gauge is then recommended to prevent these from random triggering. Thus, the gauge is sandwiched between two layers of 25 μm thick non-polarised PVDF tape, as dielectric media, and then covered with a thick aluminium adhesive tape on both side and grounded as electromagnetic shield.

Under stress, the gauge provides electrical charges that are collected by the electrical line from the PVDF sensor to the oscilloscope. The collected amount of electrical charges is not only a function of the applied pressure, polarisation and active surface but also depends on the acquisition line (cable characteristics, oscilloscope impedance, and capacitance inputs…). The collected electrical signal is interpreted knowing the calibration of the sensor and its implementation in the acquisition line. It is possible then to deduce the loading pressure.

At least three implementations of the PVDF based pressure sensor have been tested depending of the characteristic time of the mechanical loading to be measured and the acquisition chain. The “current mode” consists in measuring the voltage through a “Current Viewing Resistor” (CVR) implemented as a shunt resistor between the gauge leads. This setup is used when the rise time of the measured signal is greater than the product of the resistance *R* and capacitance *C* of the equivalent electrical circuit, including *R* and *C* values of the acquisition chain. In other cases, the PVDF gauge is implemented as “voltage mode” in which the voltage is directly measured between leads, without CVR. An alternative method for long impulses involving frequencies lower than 150 kHz is the use of a charge amplifier.

According to some preliminary numerical simulations, the expected rise time of the pressure peak in the studied case of underwater explosion is estimated to be of the order of one microsecond. The R∗C value of the circuit is estimated with the description of the acquisition line: 1 m long extended gauge using conductive copper tape, 2.5 m of RG58/U
BNC cable (102 pF/m), 15 m of CNT400 low loss cables (78 pF/m) and a digital OWON
SDS8102 oscilloscope (input 10 pF/m, 1 MΩ).

Considering the values of each portion of the acquisition chain, the product R∗C is then estimated:*R*#1 MΩ*C* # 1500 pFR∗C # 1.5 ms

The R∗C value of the acquisition chain is much greater than the rise time to be measured; then the PVDF gauge was implemented in “voltage mode”, that is to say, the oscilloscope input channel is set in 1 MΩ impedance. Considering the gauge surface of 25 mm${}^{2}$, as well as the capacitance and the resistance of the acquisition line, the voltage signal is converted into current per surface unit with the relation:(1)$$Q_{s}=V*C/S$$

$Q_{s}$ is the electric charge in Coulomb per surface unit, *V* is the voltage, *C* is the capacitance, and *S* is the active area of the PVDF gauge.

Thus, the measured pressure is deduced from relation (1) and Figure 2.

The uncertainties of measurements of the PVDF gauge in an underwater environment are deeply discussed in [18]. Another advantage of using PVDF gauges, they are movable and can also be implementable under the form of an array for field pressure measurement [12].

### 2.2. HEHF Experiments

Plastic posts are positioned normal to the steel anvil to hold detonating cords at a chosen stand-off distance (Figure 3). The explosive charge length is chosen long enough to avoid detonator and extremities effects, and to ensure that the detonation wave propagates at a constant velocity.

The acquisition chain can be triggered by two different ways. In the first one, the trigger level is adjusted to an estimated voltage deduced by the pressure which is calculated by numerical modelling or analytical consideration. This trigger option allows recording long duration until the bubble pulsation. The second one is to use the external trigger of the oscilloscope: it is directly connected to the electrical wires of the firing system that ignite the detonator. The delay between the electrical signal and the detonation time was measured to be around half a millisecond. This trigger option allows an estimation of the travel time from the explosive charge to the plate. Then, the sensor is stuck with double layer adhesive tape on the metal plate. Two sensor locations were chosen with respect to the explosive charge: (1). below, (2). with a lateral offset. The first configuration was used to determine the pressure as a function of stand-off distance vertically above the gauge and the second configuration to show the presence of a Mach stem.

### 2.3. Analytical Method

This section provides an analytical interpretation of recorded signals by considering the physics of underwater explosion and shock waves. When the detonation is triggered, a detonation wave propagates through the explosive charge. The detonation then reaches the water–explosive boundary. A shock is then transmitted into the surrounding water and propagates over the stand-off distance before striking the PVDF sensor. After a certain distance of several times the charge diameter, the shock front is assumed to be planar with respect to the active area of the sensor. The shock wave is then transmitted from the water to the aluminium shield layer of the sensor, then from the aluminium shield to the PVDF core, then to aluminium shielding of the other side, from the shield to the double-sided adhesive tape and finally to the metal plate to be formed. This situation is represented in Figure 4 showing a space-time diagram of the wave propagation in the sensor assembly (without the double sided adhesive tape) with corresponding states in a ‘Pressure–material velocity’ diagram detailed in what follows.

The simplified 1D analytical approach based on the theory of shocks in a condensed phase, described in [19], is given. In order to minimise the effects of hydrodynamic attenuation during the shock wave transit through surrounding water, only the interaction with the PVDF sensor, assuming a planar impact of the shock front on the gauge, is considered. In that case, the sensor is considered stuck on a steel plate. The assumption of a constant pressure step equivalent in impulse to the real pressure loading is made. This assumption makes possible the use of the shock polar technique: Pressure-particle velocity (P-u) diagram gives the accessible states of a non-reactive material under shock load, which is obtained by using the Rankine-Hugoniot equations [19]. Thus, the shock polar equation of a material of interest is given by the relation:(2)$$P=\rho_{0}c_{0}\left(u_{1}-u_{0}\right)+\rho_{0}s{\left(u_{1}-u_{0}\right)}^{2}$$
where *P* is comparable to the spherical content of the stress tensor in the material, $\rho_{0}$ is the material density, $c_{0}$ is the material bulk sound velocity, *s* is a material parameter given in the shock equation of state (EoS) Equation (Equation 3) with *D* the shock velocity, $u_{1}$ and $u_{0}$ are, respectively, the particle velocity under shock and at initial state (at rest).
(3)$$D=c_{0}+s\left(u_{1}-u_{0}\right)$$

The phenomenon can be better understood when described in a space-time diagram (Figure 4) and in a pressure-material velocity diagram (Figure 5). It is first assumed that PVDF and tape have the same acoustic impedance $Z=\rho_{0}c_{0}$ and of a negligible thickness compared to the aluminium shield. The shock coming from the water induces a pressure loading when impacting the sensor assembly stuck on the metal plate. During this interaction, the shock is reflected into the water and transmitted into, first, in the aluminium layer of the sensor assembly. As a consequence, the pressure state in the water and the aluminium is marked as state n°2, it corresponds to the intersection of their respective shock polar in the ‘pressure-material velocity’ diagram.

It can be seen, in such a construction, that the series of reflected waves in such assembly converge towards the state that would be without the mounted sensor. In other words, the presence of PVDF has a limited impact on the maximum pressure measurement with respect of the loading duration in this case. According to this observation, it is no longer needed to consider PVDF or adhesive tape in the shock polar approach. This will simplify the understanding.

In such a situation, the shock transmitted to the aluminium shield is then transmitted to the steel anvil and reflected in the aluminium shield. The resulting pressure states are given in Figure 5. The two materials then reach state noted n°3 (Figure 5). This pressure is the pressure measured by the gauge.

This observation means that the pressure measured by the PVDF gauge is not directly the incident pressure induced by the explosion. In order to evaluate this pressure, it is then necessary to come back again to Figure 5. The pressure states corresponding to intersection points can be found by solving the equations of the shock polars. To achieve it, a Matlab^®^ numerical file was coded; it calculates the value of the incident pressure knowing the value of the measured pressure peak. Indeed, state 3 results from the interaction of a shock coming from aluminium shield and transmitted into steel. According to the planar shock interaction, pressure induced by the shock will be equal in steel and aluminium.
(4)$$P_{steel_{3}}=P_{al_{3}}$$

$P_{steel}$ and $P_{al}$ are, respectively, the pressure of steel and aluminium under shock at state 3. Then, by using the equation of shock polar Equation (Equation 2), the following equations are obtained:(5)$$P_{steel_{3}}=\rho_{0_{steel}}c_{0_{steel}}u_{3}+\rho_{0_{steel}}s_{steel}u_{3}^{2}$$
(6)$$\rho_{0_{steel}}c_{0_{steel}}u_{3}+\rho_{0_{steel}}s_{steel}u_{3}^{2}=\rho_{0_{al}}c_{0_{al}}\left(2u_{2}-u_{3}\right)+\rho_{0_{al}}s_{al}{\left(2u_{2}-u_{3}\right)}^{2}$$

$\rho_{0_{steel}}$, $\rho_{0_{al}}$,$c_{0_{steel}}$, $c_{0_{al}}$, $s_{steel}$ and $s_{al}$ are, respectively, the density, sound velocity, and shock material parameters of steel and aluminium, $u_{3}$ and $u_{2}$ are, respectively, the particle velocity at state 3 and state 2. The steel pressure under shock has previously been determined, thus $u_{3}$ can be calculated with relation (5) or evaluated with Figure 5. Then, knowing $u_{3}$, the particle velocity at state 2 $u_{2}$ can be determined with relation (6). Thus, aluminium pressure at state 2 can be calculated with the equation of shock Equation (Equation 2), and then same work is performed with state 2 instead of state 3. Thus, $u_{1}$ can finally be obtained and the incident pressure can be found with shock polar equation in Figure 6. Common isentropic approximation is made here, atmospheric pressure is neglected, and reflected waves are supposed to follow symmetric of Hugoniot in P-u plane. It is an usual first order approximation to simplify calculation [20,21].

Thus, during a shock/plate interaction, from a PVDF pressure measurement on a material such as steel, it is possible to determine the incident pressure at a given distance from the explosive charge, and from this incident pressure it is possible to deduce what would be the pressure transmitted in any other material such as an aluminium alloy, for example. Since state 2 does not depend on the nature of the anvil, this method allows measuring the pressure loading for anvils made of any kind of materials.

## 3. Results

### 3.1. Explosive Charge Directly above the Sensor

Tests were carried out with an explosive cord positioned directly above the sensor stuck on either a steel or an aluminium anvil. The steel anvil configuration allowed performing about up to 30 trials before the gauge got lost in the pool. The explosive was placed successively above the sensor from 40 to 100 charge radii with step of 10 charge radii according to the possible precision of the experimental setup that was estimated to be ±3 charge radii. The shock wave interacting with the anvil is generated by the detonating cord, which is actually a 3D problem, simplified by the analytical 1D approach presented in Section 2.3. The measurements obtained by considering the peak of over-pressure as a function of the working distance are exhibited in Figure 7.

The repeatability of the measurement was assessed by fixing the distance at 40 times the charge radii between the detonating cord and the sensor, with identical detonating cords. A deviation between the obtained measurements can be noticed, depending on various elements: first of all, the type of double sided adhesive tape used to fix the sensor onto the anvil, seems to have a significant influence, as shown in Figure 8). Two different adhesive tapes were used, one of 30 μm, the other of 300 μm. The thickness of the thicker tape is anyway smaller than the total thickness of the sensor.

Considering the analytical approach, the first pressure peak recorded by the sensor is explained as the pressure transmitted to the sensor external aluminium shield by the shock wave coming from the water. The second peak can be considered as the shock transmitted to the steel anvil from the sensor internal aluminium shield. The P-u diagram allows deducing the value of the first peak, the materials to be considered are steel and aluminium, with the method exposed in Section 2.3. The material parameters used are given in Table 1.

The measured pressures are in good agreement with the ones obtained by analytical approach, described in Section 2.3. The relative difference with respect to the calculated pressure by analytical approach with aluminium is lower than 6.5% in the worst case at 40 charge radii (Table 2).

Figure 9 compares the role played by the nature of the anvil. The detonating cord is placed at 100 times the charge radii. As explained in Figure 5, the measured pressure peak is higher in the case of the steel anvil than for that of aluminium. Looking at a longer time, a significant difference in amplitude is observed in both recorded signals (Figure 9). A negative impulse going down to −0.38 is observed in the case of the aluminium anvil while just a short and low negative peak, down to −0.05 is noted for the steel anvil. The large negative impulse observed with the Al anvil can be explained by the fact that Al is less rigid and will bend more than steel and the PVDF sensor will be thus excited by in plane strain. Additional electrical charges will then be induced by bending. The PVDF calibration is only given for out of plane strain and as a consequence, the measured signal after the shock breaks out is disturbed when the bending is acting and cannot be considered as an applied pressure on the front of the Al anvil. An additional observation underlines the role of bending during shots with aluminium anvil. This effect was also reported by [22] in soft impact configuration.

Since steel is less subjected to bending with the same loading, the bending effects are less visible. The short and low negative pressure observed in the case of the steel anvil corresponds to some likely bubbles oscillations when stricken by the shock wave. Additionally, since the acoustic impedance of steel is higher than that of water, release waves are reflected at the back face of the anvil and come back to the sensor where they cross the release waves of the unloading. This situation may cause locally tensile loading in water. As a consequence, cavitation bubbles may appear and oscillate.

Four shots carried out on aluminium anvil led to torn off the sensor. Due to this effect, the recorded signals were of a poor quality, only one was acceptable. Shots were performed with aluminium anvil clamped on the steel anvil (Figure 10). For some other shots, the aluminium anvil was hung in water by steel hooks. This variety of boundary conditions strongly influence the anvil response to an impulsive loading. Since the acoustic impedance of aluminium is lower than that of steel and higher than the one of water, the reflected wave from aluminium to water is a tensile wave. The configurations shown in Figure 10 are also partially representative of the different stages of an aluminium plate forming: the plate is first subjected to UNDEX, supported by a mould on its edges, and its back surface is free due to vacuum between the plate and the mould, which is similar to configuration Figure 10b. Then, the plate is accelerated towards the mould and impacts it, that can be approximated with configurations Figure 10a,c. In configuration Figure 10a, the plate-mould impact is measured directly with a pressure sensor located in vis-a-vis of the place where the mould got impacted, without considering incident angle. Configuration Figure 10c represents a measurement of reflected shock from impact using a pressure sensor on the top of the aluminium plate. Configurations Figure 10b,c have to be enhanced to ensure the integrity of the pressure sensor and a more accurate measurements.

### 3.2. Angle of Incidence Effects

The study of the recorded pressure as a function of the angle of incidence of the shock is a key parameter to predict the mechanical loading on the anvil during the forming process. Nevertheless, the experimental set-up only permits an approximate measurements of the obliquity: a difference of few millimetres can induce a visible angle variation due to the proximity of the explosive charge to the anvil. Moreover, making an extensive study would be costly: a large number of shots is necessary for exploring all configurations.

A preliminary attempt remains essential to evaluate the possibility of observing the presence of a Mach stem (Figure 11). The Mach stem is a process that appears when the shock wave interacts with a rigid plate, away of the projection of the detonation point on the plate. At a critical obliquity, the combination of the reflected wave and the incident one creates an irregular state called a Mach stem. It induces a triple point between incident and reflected waves that results in an increased overpressure compared with a regular reflection. This overpressure has to be considered for a better modelling of the metal forming. It is therefore necessary to verify the role played by the Mach-stem during HEHF explosions. In addition, a better knowledge of this phenomenon may make possible to improve the experimental setup to protect the sensor.

During the experimental campaign at the 3DMF manufacture plant, measurements are performed at 50 charge radii above the steel anvil. The cord is moved away from the sensor, in order to get measurements at 15${}^{\circ }$, 30${}^{\circ }$, 45${}^{\circ }$, and 60${}^{\circ }$ of angle of incidence, see Figure 11 and Table 3.

A pressure jump is observed between 30${}^{\circ }$ and 45${}^{\circ }$. This confirms the formation of a Mach stem in the water. The triple point (combination of the incident and reflected waves) appears around 40° as it is reported for explosive shocks in air [23].

## 4. Discussion

In order to estimate the relevance of using PVDF shock gauges for dynamic pressure measurement during underwater explosion of a detonating cord, refined numerical modelling were performed. Two commercial computational codes have been used in this study: SPEED^®^ and RADIOSS^®^. Shock physics explicit Eulerian/Lagrangian dynamics (SPEED^®^) is developed by Numerics Gmbh, and has capability to perform 2D and 3D multi-material Eulerian simulations, 2D Lagrangian models and 3D ideal gas calculations. It is based on an explicit solver for non-linear problems and is proficient for shocks and explosion analysis. In addition, an adaptive mesh, and mesh activation method are implemented and are revealed to be very efficient for saving computational time. RADIOSS^®^ is an ALTAIR^®^ code based on a finite element method using an explicit solver, mainly for crashes and multi-physics analysis. It has multiple possibilities such as Eulerian, Lagrangian, Arbitrary Langrangian Eulerian (ALE), Coupled Eulerian-Lagrangian (CEL) and Smooth Particles Hydrodynamics (SPH) simulations in 2D and 3D.

Firstly, to ensure the accurate modelling of underwater explosion, mesh sensitivity is studied. Then, the full PVDF sensor is modelled and the pressure impulse is compared with the measurement. Finally, the correlation between measured overpressure from 40 to 100 radii from the centre of the explosive and the simulation is attempted.

For the following simulations the detonation point is set at the centre of the explosive and all the borders of the domain are set as transmissive. Transmissive boundary conditions permit the flow to go in and out of the mesh, and allow to model continuous flow. It is aimed to limit shock reflection on the boundaries. The constitutive law used for PVDF and water is SPEED^®^ shock EoS based on Mie-Grüneisen and shock EoS. SPEED^®^ Johnson-Cook constitutive law (a combination of shock EoS and visco-elasto-plastic and fracture Johnson-Cook models) is used for aluminium (AA2024-T351) and steel (Steel 1006) [24,25]. The material parameters for shock EoS are mentioned in Table 1—for aluminium, steel and water—and in Table 4 for PVDF. The explosive is modelled using the Jones-Wilkins-Lee (JWL) equation of state [26]. The JWL equation of state corresponds to the equation of state of detonation products. It is an usual formulation for dealing with detonation by numerical modelling, implemented in numerous software, under this formula:(7)$$P=A\left(1-\frac{\omega }{R_{1}V}\right)e^{-R_{1}V}+B\left(1-\frac{\omega }{R_{2}V}\right)e^{-R_{2}V}+\omega \frac{E}{V}$$

*A*,*B*,ω,$R_{1}$, and $R_{2}$ are JWL parameters depending on the explosive, *E* is the energetic material energy. V is defined here as the ratio between the initial explosive density $\rho_{0}$ and the detonation products density ρ. These parameters govern the expansion of the bubble induced by expansion of detonation products. For confidential reasons, the set of JWL parameters is not given.

In SPEED^®^, the EoS is defined by a linear relation used to calculate the Hugoniot pressure with the shock Equation (Equation 3), as explained in SPEED^®^ reference guide:(8)$$P_{H}-P_{ref}=\frac{\rho_{ref}c_{0}^{2}\eta }{{\left(1-s\eta \right)}^{2}}$$
(9)$$\eta =1-\frac{\rho_{ref}}{\rho }$$
$P_{H}$ is the pressure on the shock Hugoniot depending on the density, such as: $P_{H}$
$\left(\rho_{ref}\right)=P_{ref}$. ρ and $\rho_{ref}$ are, respectively, the shocked material density and the reference material density. Then the Hugoniot internal energy $e_{H}$ is obtained knowing the reference internal energy $e_{ref}$:(10)$$e_{H}\left(\rho \right)=e_{ref}+\frac{1}{2}\left(P_{H}\left(\rho \right)+P_{ref}\right)\left(\frac{1}{\rho_{ref}}-\frac{1}{\rho }\right)$$

The pressure *P* at an arbitrary state is described as follow, with *e* the internal energy:(11)$$P\left(\rho ,e\right)=P_{H}\left(\rho \right)+\Gamma \rho \left(e-e_{H}\left(\rho \right)\right)$$
with the Grüneisen coefficient gamma Γ that depends on volume, and reference Grüneisen coefficient $\Gamma_{0}$: (12)$$\Gamma =\frac{\Gamma_{0}\rho_{ref}}{\rho }$$

In addition to that, if $\mu =\frac{\rho }{rho_{ref}}-1<0$, the pressure is defined as:(13)$$P\left(\rho ,e\right)=\rho_{ref}c_{0}^{2}\mu +\Gamma_{0}\rho_{ref}e$$

The temperature curve reference used to calculate the temperature is the Hugoniot:(14)$$T\left(\frac{1}{\rho },e\right)=T_{H}\left(\frac{1}{\rho }\right)+\frac{1}{c_{v}}\left(e-e_{H}\left(\frac{1}{\rho }\right)\right)$$

*T* is the material temperature, $T_{H}$ is the Hugoniot temperature, $c_{v}$ is the specific heat capacity at constant volume.

$T_{H}$ is defined as the solution of the following differential equation, where ν is the specific volume:(15)$$\frac{dT_{H}}{d\nu }+\frac{\Gamma }{\nu }T_{H}=\frac{1}{c_{v}}\left(P_{H}+\frac{de_{H}}{d\nu }\right)$$

In RADIOSS^®^, the hydrodynamic pressure *P* of an element can be calculated using a polynomial expression as explained in its reference guide:(16)$$P=C_{0}+C_{1}\mu +C_{2}\mu^{2}+C_{3}\mu^{3}+\left(C_{4}+C_{5}\mu \right)E_{V0}$$

Here $\mu =1-\rho /\rho_{ref}$, $E_{V0}$ is the internal energy per volume unit. For condensed phase, the cubic Hugoniot Equation (Equation 17), Mie-Grüneisen Equation (Equation 11) and energy conservation Equation (Equation 10) equations are used:(17)$$P_{H}=\rho_{ref}c_{0}^{2}\mu +\rho_{ref}c_{0}^{2}\left(2s-1\right)\mu^{2}+\rho_{ref}c_{0}^{2}\left(3s-1\right)\left(s-1\right)\mu^{3}$$

By substituting Equations (17) and (10) in Equation (Equation 11), polynomial coefficients can be identified as:(18)$$C_{0}=P_{ref}$$
(19)$$C_{1}=\rho_{ref}c_{0}^{2}-\frac{\Gamma_{0}}{2}P_{ref}$$
(20)$$C_{2}=\rho_{ref}c_{0}^{2}\left(2s-1\right)-\frac{\Gamma_{0}}{2}\rho_{ref}c_{0}^{2}$$
(21)$$C_{3}=\rho_{ref}c_{0}^{2}\left(3s-1\right)\left(s-1\right)-\frac{\Gamma_{0}}{2}\rho_{ref}c_{0}^{2}\left(2s-1\right)$$
(22)$$C_{4}=C_{5}=\Gamma_{0}$$

The Johnson-Cook model gives the material flow stress *Y* depending on strain rate, strain and temperature:(23)$$Y=\left(A+B\epsilon_{p}^{n}\right)\left(1+Cln\left(\frac{\dot{\epsilon }}{\dot{\epsilon_{0}}}\right)\right)(1-\left({\left(\frac{T-T_{room}}{T_{melt}-T_{room}}\right)}^{m}\right)$$

*A*, *B*, *C*, *n* and *m* are Johnson-Cook coefficients. $\epsilon_{p}^{n}$ is the plastic strain, $\dot{\epsilon }$ and $\dot{\epsilon_{0}}$ are, respectively, the strain rate and reference strain rate, *T* and $T_{melt}$ are, respectively, the material temperature and melt temperature. $T_{room}$ is the room temperature.

The mesh convergence study is performed with SPEED^®^ software. An explosive cord is placed in the centre of a 90 charge radii square computational domain of water and a numerical gauge is put at 40 charge radii from the centre of the explosive (Figure 12). All mesh elements are squares. The mesh sensitivity of the model is studied and the element size was fixed to 2 mm for the coarser model, to 0.05 mm for the finest one. The pressure history during wave propagation from the detonating cord is monitored (Figure 13) by implementing numerical sensors placed in the mesh.

The maximum pressure observed with a 2 mm and 1 mm element size mesh are about the same: 0.22. However, the impulse with an element size of 2 mm is greater. By lowering the element size to 0.5 mm, the maximum pressure is about 0.29: it is a 32% percent increase with respect to the 1 mm maximum pressure. Dividing again the element size by 2, the maximum pressure still increases up to 0.34: it is about 17% more than the 0.5 mm pressure. Even with smaller element size the pressure continues to rise to 0.4 and 0.44, respectively, for 0.1 mm and 0.05 mm element size which represents more than a 10% increase, and it is not negligible. The computed peak of pressure shows that a 0.1 mm mesh element size is still not converging to pressure obtained with a 0.05 mm element size. However, with a 2D mesh, dividing the element size by 2 means that the quantity of elements in the model is multiplied by 4. Thus, smaller mesh element size yield obviously to a better description of the over-pressure versus time, but higher computation time. Although the maximum pressure is noticeably higher, the impulse gain is not that big: depending on the case, the additional computation time could be not worthy. Moreover, the pressure is possibly not realistic when closer to the cord but could be relevant at a longer distance with larger element size, this has been described in [27].

To check the correlation between measurement and simulation, the measurement process is simulated (Figure 14). The sensor is then modelled and a numerical pressure gauge is placed at its centre. The 2D explicit simulation is performed using the SPEED^®^ software, with quad elements of size of 100 μm.

Computed and measured pressures are compared in Figure 15. Some differences are observed and for which some explanations are given. The actual problem geometry is in 3 dimensions whereas the simulation is in 2D plane (extruded). The element size might not be enough refined to obtain a realistic rise time (only one element in the thickness of PVDF in order to have a reasonable computational time). Nevertheless, there is still an acceptable correlation between maximum pressures (less than 1% of relative difference), which confirms the relevance of the measured pressure. An oscillation with high frequency is observed in the experimental measure. They are attributed to the reflected waves inside the pressure sensor layers. Its period is around 0.25 (dimensionless unit) which gives a length of 1.3 charge radii which is almost the length of the aluminium shield on one side of the sensor. The impulse difference between measurement and simulation can be explained: PVDF is very thin, thus, there are only a few elements in the thickness. The PVDF is crushed when the shock wave goes through the material under intense compression in the simulation. The mesh element does not contain any more PVDF material after few instants but the numerical sensor remains. Thus, there is lost energy during the interaction between materials. Furthermore, the mesh element size is surely not small enough to represent accurately the PVDF behaviour.

From the measurements obtained with one detonating cord and analytical treatment of shock waves in condensed phase, the pressure applied on the sensor can be deduced. A 2 dimensions model is computed with SPEED^®^ and RADIOSS^®^: an explosive is set into a 210 charge radii square domain of water with numerical gauges spaced every 5 charge radii (cf. Figure 16). The goal is to compute the peak of overpressure and to compare it with measurements with the analytical method presented in Section 2.3 to get the incident pressure in a range of 40 to 100 charge radii (cf. Figure 5 and Figure 17).

The relative difference between RADIOSS^®^ and SPEED^®^ with the measurement is calculated at each distance with respect to measurement values in Table 5. The relative difference is mainly under 10% up to 90 radii except for one shot at 40 radii. The computed pressure tends to be higher than the measured pressure under 60 radii and under the measured pressure from 70 radii away from the explosive. However, this statement has to be taken carefully since not many shots were performed.

The differences between experiments and simulations can be explained mainly by the element size and numerical dissipation. Indeed, as previously stated, the peak of overpressure continued to increase with the decrease in element size. In order to have a reasonable computation time (less than 24 h with a desktop computer), the size of the elements was set, respectively, to 0.25 mm and 0.05 mm in RADIOSS^®^ and in SPEED^®^ models. The smaller element size obtained with SPEED^®^ could be achieved by using the mesh activation feature, saving a lot of CPU time. Both software give similar trends, and are close to the estimated incident pressure from experimental data, except for the longest distances (Figure 17). An explanation could be that the pressure of elements cannot reach $P_{CJ}$ (Chapman-Jouguet pressure) in the explosive during the detonation stage and could imply not only a reduction in pressure at the source but also a dissipation of energy due to the phenomena of numerical dissipation (Figure 13) for the longest distances. The method for estimating the incident pressure could also contribute to discrepancies since it is based on the 1D approximation.

The pressure measurement of underwater explosion using a PVDF based pressure sensor gave realistic results. However, the gauge remains intrusive and multiple shock reflections within its constitutive layers can be noticed on the measurements. It has been discussed that both experiments and numerical simulations cannot correlate perfectly due to the intrusiveness of the sensor and assumptions made for the numerical modelling. Nevertheless, the amplitude of shock pressure is overestimated by the numerical modelling while the stand-off distance remains below 65 charge radii and under estimated further. Largest discrepancies of calculated incident pressure with respect to the measurement remain around 15%. The first phenomenon observed is a large discrepancy on impulse, mainly due to insufficiently small mesh size. Thus, energy loss appears in computation and will results in large differences in forming prediction.

## 5. Conclusions

High explosive hydro forming by detonating cord is identified to be a promising technique for the forming of large and thick metal plates (several meters). However, its result depends particularly on the ability of tuning and measuring the mechanical loading exerted on the plate to be formed. However, no analytical nor empirical laws have been found in the open literature for the description of the shock pressure generated by underwater detonating cords for stand off distances used in HEHF. In order to complete the description of high explosive hydro forming of large metal plates, a pressure sensor, based on a PVDF shock gauge is developed as a low-cost technique. In this paper, it has been shown that:-The measurement method and signal interpretation is described by a physical and analytical description. It pointed out the relevance of the measured peaks of over-pressure. The pressure amplitudes given by the sensor are close to the ones deduced by the analytical approach and computed with numerical modelling. The analysis presented allowed calculating loading pressure exerted on the metal plate. Considerations on incident and reflected shocks were efficient to take into account the multi-layer PVDF gauge and its intrusiveness.-Measurements of the peaks of over-pressure versus distance correlate with two different numerical models performed with two different commercial codes based on different numerical methods.-The formation of a Mach stem at about between 30${}^{\circ }$ and 45${}^{\circ }$ propagating on the metal anvil has been evidenced in underwater environment and results in an additional mechanical loading on the fluid/metal plate interaction that has to be considered in the HEHF process.-Its implementation does not disturb the forming process of large and thick metal plates and allows an in-situ dynamic pressure monitoring during the HEHF process.-In the case of rigid plates, the sensor can be reusable several times, for about 30 shots before it broke.

These findings provide valuable model validation necessary to numerically design the HEHF process for thick metal plates prior to the manufacturing stage. This last fact allows reducing the number of preliminary explosive shot—necessary to adjust the process parameters—that are costly. They also offer the possibility of coupling the HEHF process with artificial intelligence better control and monitoring.

However, the variability of the pressure records has not been extensively evaluated, due to the limited amount of shots performed at each distance. The next step could be to evaluate the full incident impulse with transmitted measurement based on a detailed algorithm, which would consider each layer’s thickness and material. This would give a deeper understanding of the fluid-structure interaction. Additionally, more measurements (or one measurement with many gauges) in configuration of interest and explosive are needed to determine a field of pressure to be implemented in a simulation software or analytical model of plate deformation. In addition to that, experiments and simulations of a typical aluminium plate hydroforming case could be performed to evaluate the influence of the plate’s thickness, the deformation of the part on pressure signal, and the effect of having air, void, or water between plate and mould. In this case, as it was observed that the gauge got torn when on an aluminium anvil, its integrity should be enhanced to be used in more shots. On top of that, at least one sensor should be placed directly on the mould to measure pressure impact of the formed plate.

## Figures and Tables

**Figure 1 sensors-21-04429-f001:**
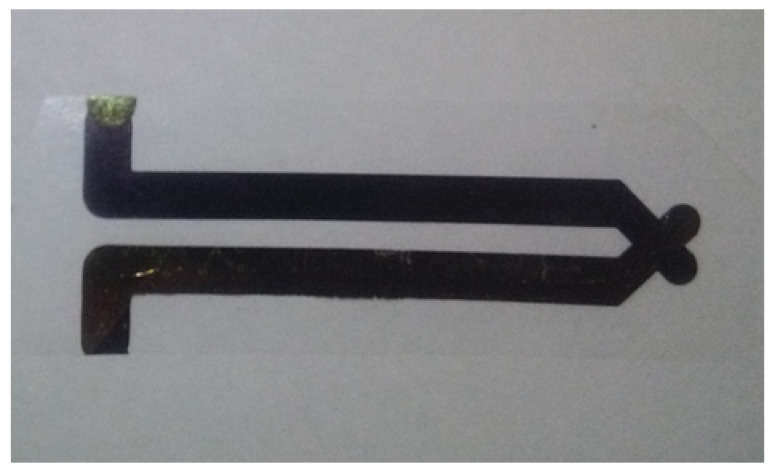
Bauer S25 PVDF bare shock gauge. Leads are on each side of the tape, sensing area is located at their facing portions.

**Figure 2 sensors-21-04429-f002:**
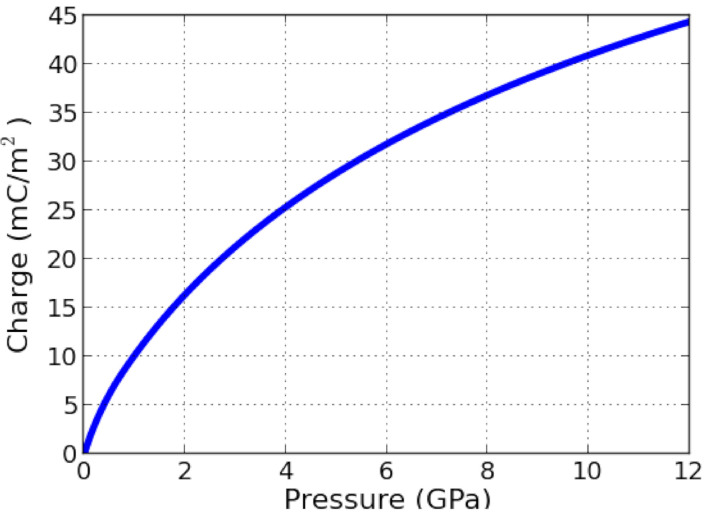
Calibration of electrical charges per unit of area produced by PVDF versus loading pressure under mono-dimensional shock pressure (inspired from [14]).

**Figure 3 sensors-21-04429-f003:**
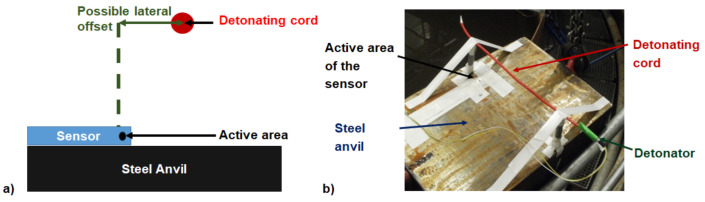
Shielded PVDF sensor stuck with tape on the steel anvil. Detonating cord positioned above with plastic posts and tape. Scheme (**a**) and picture (**b**).

**Figure 4 sensors-21-04429-f004:**
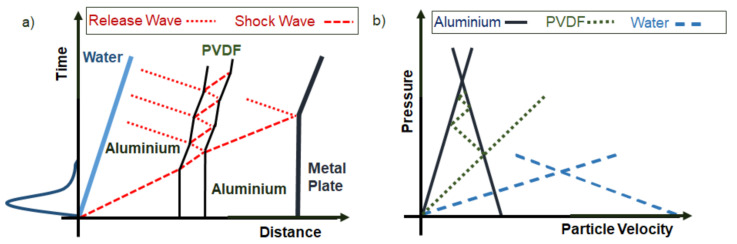
x-t (distance-time) diagram (**a**) and P-u (pressure-particle velocity) diagram (**b**) describing the interaction of an assumed planar shock with constitutive layers of the sensor. Shocks and release waves are in red. The primary shock comes from water, goes through aluminium shield, PVDF and aluminium before being reflected. The successive states reached by the PVDF are represented through the P-u diagram.

**Figure 5 sensors-21-04429-f005:**
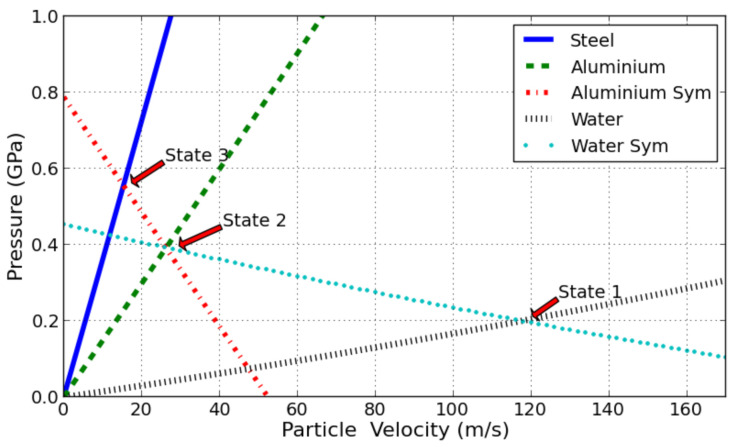
P-u diagram showing the successive states of water, aluminium, and steel under a 200 MPa incident shock load. The shock comes from the water, is transmitted into the aluminium and then into the steel.

**Figure 6 sensors-21-04429-f006:**
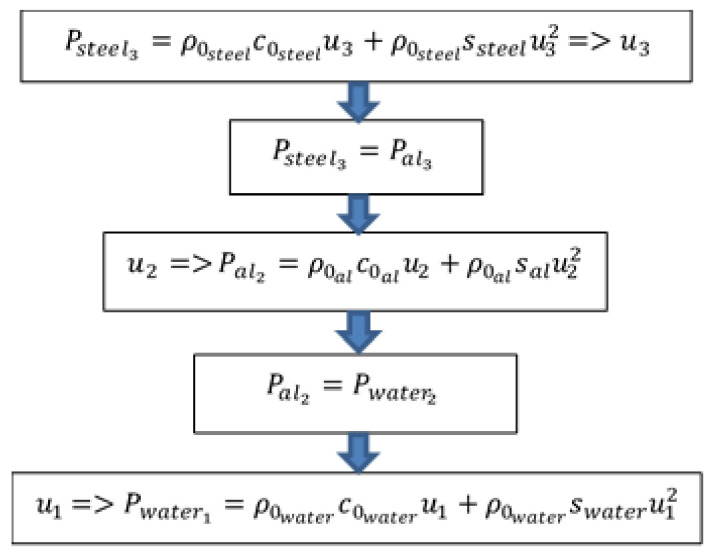
Flowchart of Matlab^®^ numerical file used to calculate the incident pressure from transmitted pressure to steel using 1D shock theory.

**Figure 7 sensors-21-04429-f007:**
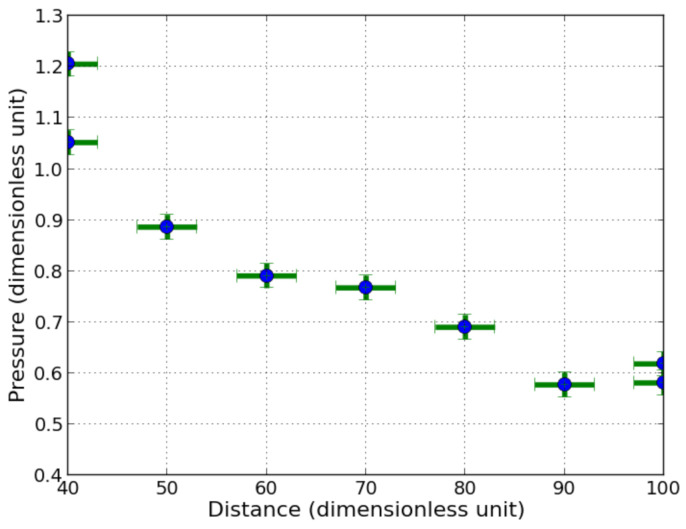
Measured peak pressures with sensor on steel anvil versus distance with one detonating cord (dimensionless units).

**Figure 8 sensors-21-04429-f008:**
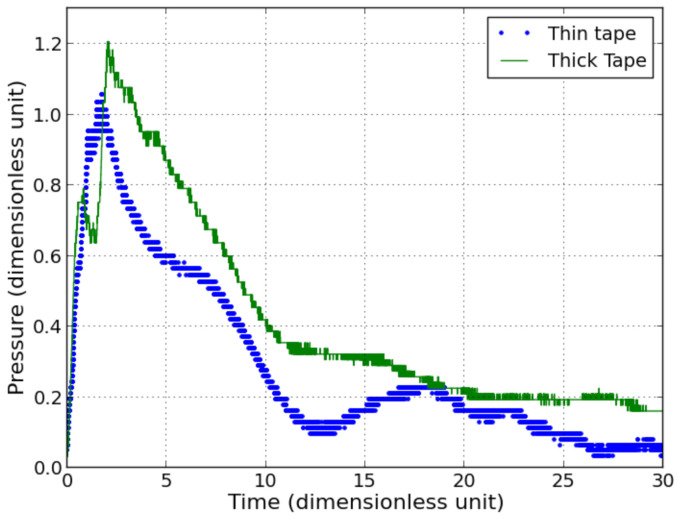
Measured pressure with sensor onto steel anvil using thin (30 μm) and thick (300 μm) double sided adhesive tape, from one detonating cord (dimensionless unit).

**Figure 9 sensors-21-04429-f009:**
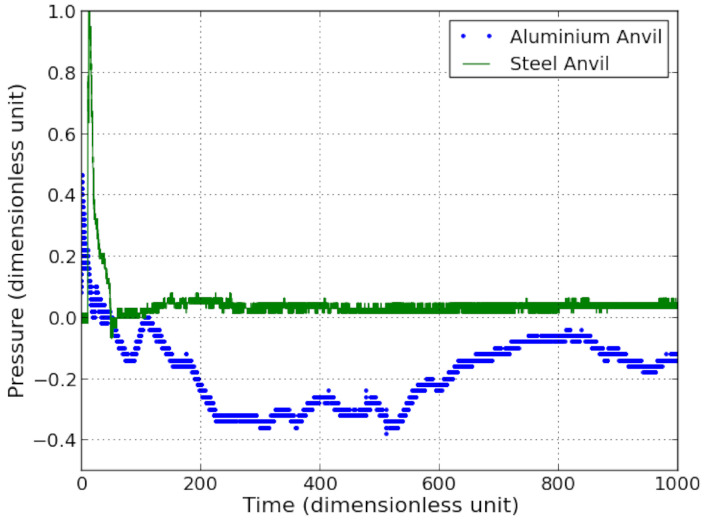
Signals obtained with a PVDF gauge (dimensionless units) on aluminium anvil and on steel anvil using explosive at 100 charge radii.

**Figure 10 sensors-21-04429-f010:**
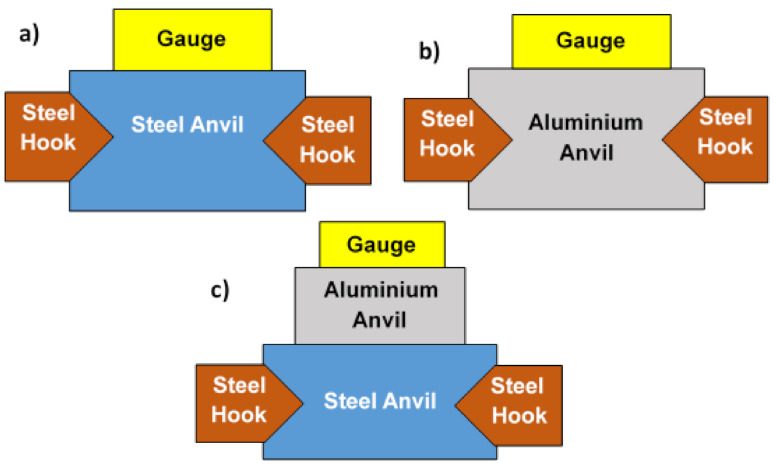
Variants of the setup: pressure sensor on steel anvil (**a**), pressure sensor on aluminium anvil attached with steel hooks (**b**), pressure sensor on aluminium anvil clamped over steel anvil (**c**).

**Figure 11 sensors-21-04429-f011:**
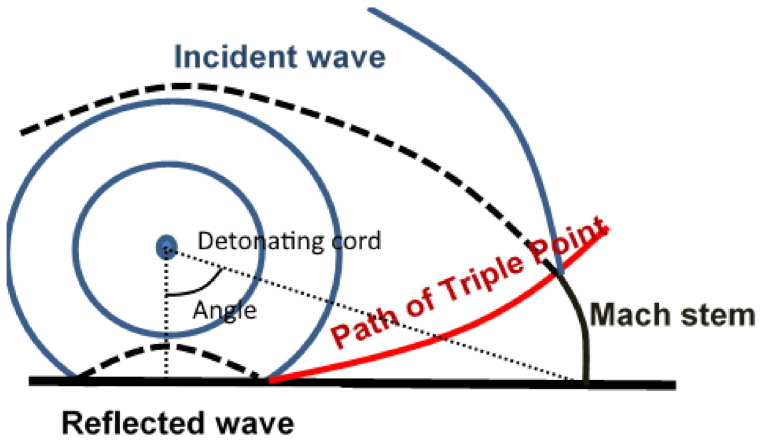
Sketch of Mach stem and triple point emergence and behaviour.

**Figure 12 sensors-21-04429-f012:**
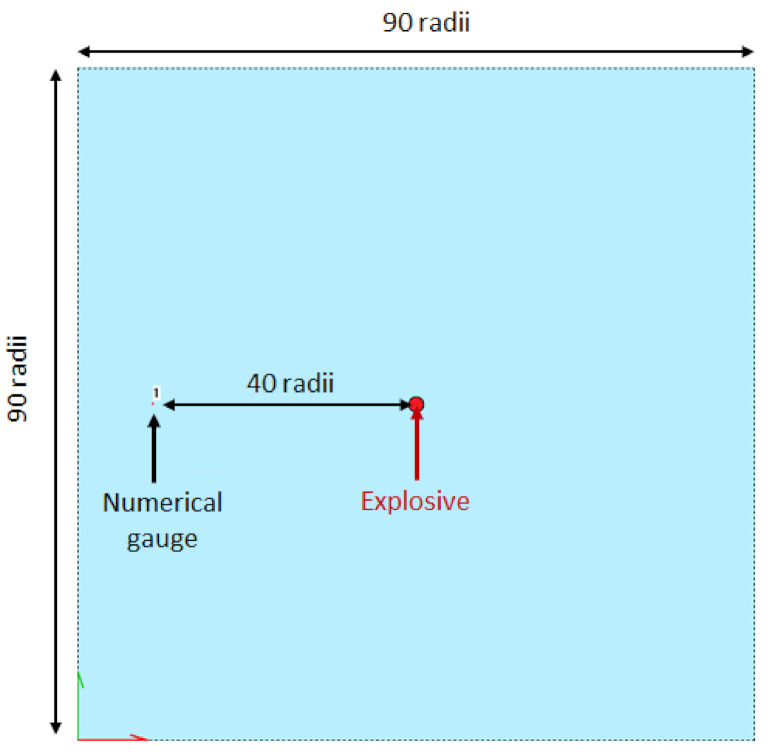
Configuration of the mesh sensitivity study simulation study performed with SPEED^®^. An explosive is set in the centre of the mesh domain with a numerical gauge at 40 radii away from it. The element size varies from 0.05 mm to 2 mm, and the domain is a 90 radii square.

**Figure 13 sensors-21-04429-f013:**
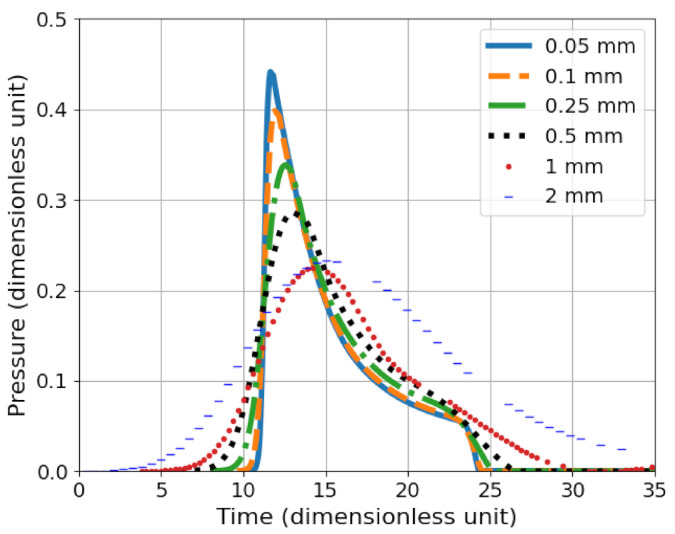
Mesh convergence study using 2D SPEED^®^ simulation (dimensionless unit), the explosive is set at 40 charge radii from the numerical gauge.

**Figure 14 sensors-21-04429-f014:**
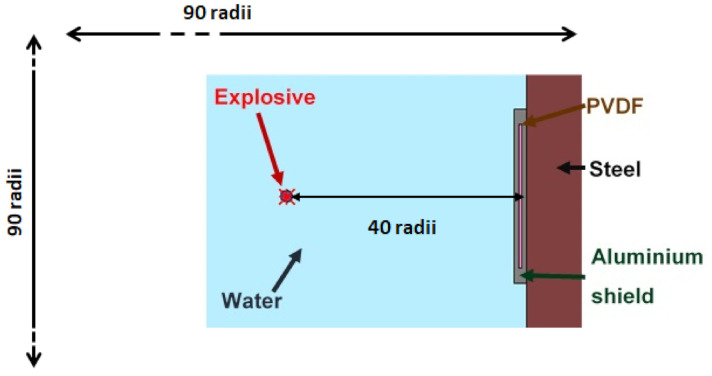
SPEED^®^ modelling of detonating cord underwater explosion above PVDF gauge placed on steel anvil.

**Figure 15 sensors-21-04429-f015:**
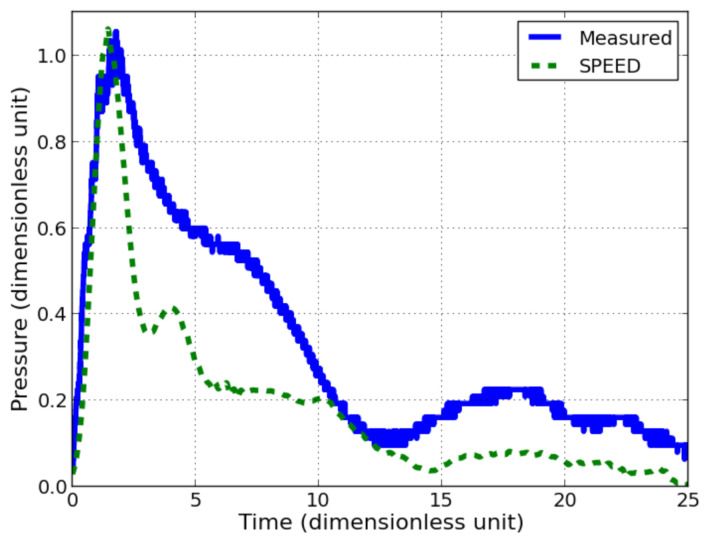
Dimensionless measured pressure with pressure sensor on steel anvil with an explosive at 40 charge radii and SPEED^®^ recored signal in PVDF with equivalent simulation, with 0.1 mm element size.

**Figure 16 sensors-21-04429-f016:**
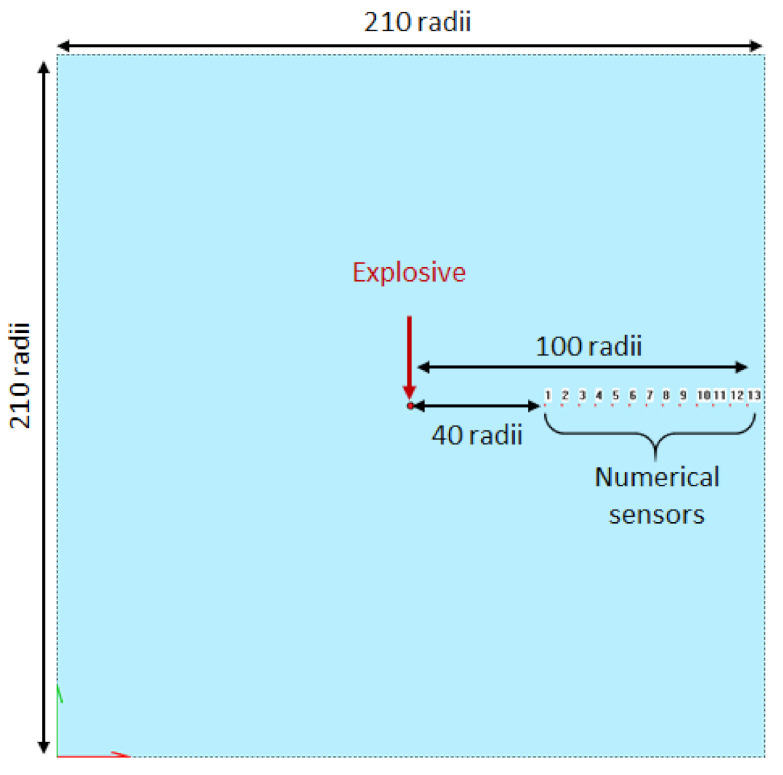
Configuration of the underwater explosion simulation of a detonating cord. The explosive is set in the centre of the mesh domain with numerical gauges from 40 to 100 radii away from it with a 5 radii step. The element size is equal to 0.05 mm with SPEED^®^ and 0.25 mm with RADIOSS^®^, and the domain is a 210 radii square.

**Figure 17 sensors-21-04429-f017:**
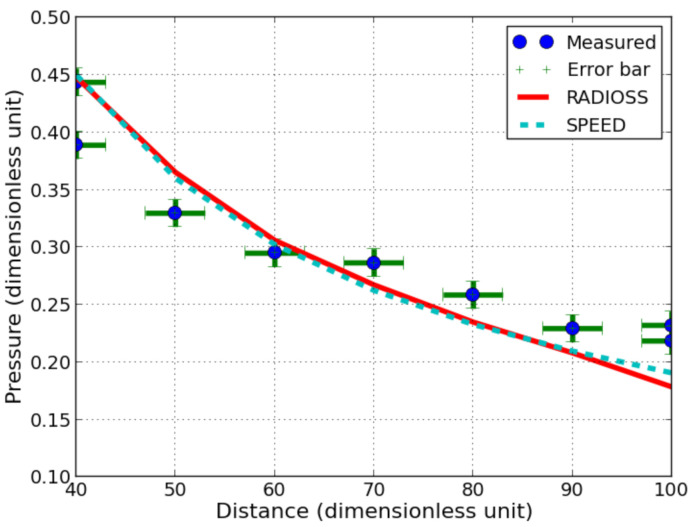
Comparison of incident pressure versus distance between experiments and simulation with RADIOSS^®^ (0.25 mm square elements) and SPEED^®^ (0.05 mm square elements) using dimensionless units. Experimental pressure is converted to incident pressure using 1D shock theory as explained in Section 2.3.

**Table 1 sensors-21-04429-t001:** Steel, aluminium and water shock equation of state parameters from [19] used to plot shock polar in Figure 5.

Material	Density (kg/m^3^)	Sound Velocity (m/s)	s
Steel	7896	4569	1.490
Aluminium	2785	5328	1.338
Water	998	1483	1.921

**Table 2 sensors-21-04429-t002:** Dimensionless peaks of overpressure measured by the sensor at various distances. The highest peak pressure corresponds to interaction between sensor aluminium shield and steel and the first peak corresponds to the interaction between sensor aluminium shield and water. The relative difference is given with respect to the highest peak.

Distance	Highest Peak Pressure Al/Steel	First Peak Pressure Al/Water	Pressure in Al by Analytical Approach	Rel. Diff.
40	1.210	0.791	0.845	6.4%
50	0.886	0.644	0.622	3.5%
60	0.791	0.548	0.544	1.1%
70	0.767	0.512	0.538	4.8%
80	0.690	0.477	0.484	1.4%
90	0.578	0.387	0.406	4.7%
100	0.581	0.387	0.408	5.1%

**Table 3 sensors-21-04429-t003:** Dimensionless measured peak pressure with various angles of incidence (see Figure 11) at 50 charge radii above steel anvil.

Angle	0°	15°	30°	45°	60°
Pressure	0.886	0.724	0.544	0.640	0.476

**Table 4 sensors-21-04429-t004:** Material parameters of PVDF required to model underwater explosion based on shock Equation of State [19].

Material	Density (kg/m^3^)	Sound Vel (m/s)	s
PVDF	1767	2579	1.586

**Table 5 sensors-21-04429-t005:** Relative difference (%) between the peak of overpressure from simulation with SPEED^®^ and RADIOSS^®^ simulation with incident pressure calculated with method presented in Section 2.3 and measurement (Figure 7). Since 2 shots were performed at 40 and 100 radii, both are presented with a specific label namely “(1)” and “(2)”. The percentage is computed with respect to measured values.

Distance	40 (1)	40 (2)	50	60	70	80	90	100 (1)	100 (2)
SPEED^®^	−1.2	−15.5	−6.2	−1.8	8.2	9.2	8.4	13	18.1
RADIOSS^®^	−1	−15.2	−10.8	−3.8	6.7	9.1	9.2	18.3	23.1

## Data Availability

Data supporting reported results can be found by demand to the corresponding author.

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
