# Peer review of "PVDF Based Pressure Sensor for the Characterisation of the Mechanical Loading during High Explosive Hydro Forming of Metal Plates"

_sensors, 2021, doi:10.3390/s21134429_

Round 1
Reviewer 1 Report
Dear author,
The paper has a very high quality. It is a good source for those who seek technical information on numerically modeling UNDEX, particularly for HEHF.
I would just advise you to update and enrich the state-of-the-art review of the Introduction section. I know that, particularly for the simulation part, this is a field in ongoing evolution and it should not be hard for you to find recent references. This would demonstrate that your research is a current trending topic and that the scope is in line with the latest findings, thus justifying it.
Author Response
Autjors want to thank reviewer 1 for the time he/she spent for reviewing our article.
In bold comments made by reviewer 1 :
"The paper has a very high quality. It is a good source for those who seek technical information on numerically modeling UNDEX, particularly for HEHF."
Authors thank reviwer 1 for these comments, we are please to know that you appreciated our work.
"I would just advise you to update and enrich the state-of-the-art review of the Introduction section. I know that, particularly for the simulation part, this is a field in ongoing evolution and it should not be hard for you to find recent references. This would demonstrate that your research is a current trending topic and that the scope is in line with the latest findings, thus justifying it."
Authors thank the reviwever for this relevent advice. We have added at least two recent references in the following paragraph : "Furthermore, when developing predictive numerical modelling, engineers are coping with a computation time that is dependant of the complexity of the brief and intense phenomenon to be considered, especially for underwater detonation related studies [4],[5]. For instance, high explosive detonation has a characteristic time below the microsecond, then the underwater shock that reaches the plate in few tens of microseconds and creates a fluid - structure interaction that is not always well implemented in numerical46codes and increases considerably the computational time [6]. "
with :
4.VANNUCCHI DE CAMARGO, F.; Survey on experimental and numerical approaches to model underwater explosions.Journal ofMarine Science and Engineering, 7(1), 15(2019).
5.CAO, L., FEI, W., GROSSHANS, H., and CAO, N. ; Simulation of underwater explosions initiated by high-pressure gas bubblesof various initial shapes.Applied Sciences, 7(9), 880(2017)
They are the most recent we could find in a short response time imposed by the editor. We hope they will enhance the content of our work.
Reviewer 2 Report
In my opinion the paper should be accepted - just do final spell check
Author Response
Authors want to thank reviewer 2 for his/her good appreciation of our work.
We did a final spell check. We found a unappropriate legend on fig. 13, it has been modified.
Line 429 "Considering the detonation stage and the fluid-structure interaction is considerably time consuming in term of computation." was quite confuse and out of context at this place so we have deleted it.
Other minor spell checks are also reported in the final version.
Reviewer 3 Report
The authors designed the PVDF pressure sensor for the HEHF technique. The paper is interesting. But some questions should be answered:
- In the background, large and thick plates are aimed, but the simulation case is a simple and 2D one;
- The presentation of simulation results is poor. The majority of the useful results are lacking;
- The simulation settings are unclear;
- How to consider the failure of the pressure sensors?
Round 2
Reviewer 3 Report
The authors have answered all questions in detail. The paper now is good and it is recommended to publish.